# Bovine Viral Diarrhea Virus: Recent Findings about Its Occurrence in Pigs

**DOI:** 10.3390/v12060600

**Published:** 2020-05-31

**Authors:** Luís Guilherme de Oliveira, Marina L. Mechler-Dreibi, Henrique M. S. Almeida, Igor R. H. Gatto

**Affiliations:** 1School of Agricultural and Veterinarian Sciences, São Paulo State University (Unesp), Jaboticabal. Via de Acesso Prof. Paulo Donato Castelanne s/n, Jaboticabal - SP 14884-900, Brazil; mlopesvet@gmail.com (M.L.M.-D.); henri_almeida2003@yahoo.com.br (H.M.S.A.); 2Ourofino Animal Health Ltda. Rodovia Anhanguera SP 330, Km 298. Distrito Industrial, Cravinhos – SP 14140-000, Brazil; igatto_10@hotmail.com

**Keywords:** BVDV, experimental infection, natural infection, pigs

## Abstract

Bovine viral diarrhea virus (BVDV) is an important pathogen belonging to the *Pestivirus* genus, *Flaviviridae* family, which comprises viral species that causes an economic impact in animal production. Cattle are the natural host of BVDV and the main source of infection for pigs and other animal species. Due to its antigenic and genetic similarity with other important pestiviruses such as Classical Swine Fever Virus (CSFV), several studies have been conducted to elucidate the real role of this virus in piglets, sows, and boars, not only in the field but also in experimental infections, which will be discussed in this paper. Although BVDV does not pose a threat to pigs as it does to ruminants, the occurrence of clinical signs is variable and may depend on several factors. Therefore, this study presents a survey of data on BVDV infection in pigs, comparing information on prevalence in different countries and the results of experimental infections to understand this type of infection in pigs better.

## 1. Updates on BVDV Infection in Swine

Bovine viral diarrhea (BVD) is an infection caused by the bovine viral diarrhea virus (BVDV), belonging to the genus *Pestivirus*, family *Flaviviridae*, with single-stranded positive polarity RNA [1]. Viruses belonging to the *Pestivirus* genus infect hosts of several animal species and include viral agents of great impact for animal production [2,3]. The *Pestivirus* species have been recently named from A to K and, among them, the *Pestivirus* A (BVDV-1), *Pestivirus* B (BVDV-2), *Pestivirus* C (Classical Swine Fever Virus), and *Pestivirus* K (atypical porcine pestivirus) are the main viral species related to swine [4]. BVDV has two genotypes, type 1 and type 2, which are classified into sub-genotypes: BVDV-1 (1a to 1u), adding up to 21 sub-genotypes, and BVDV-2 (2a to 2d), with four sub-genotypes [5]. BVDV-1 is related to most reference strains, is commonly used for vaccine production, and was most frequently isolated from mild to moderate clinical cases in cattle. Conversely, BVDV-2 was isolated from acute disease outbreaks, also presenting strains of mild and moderate virulence [6].

Based on the effect of replication on cell culture, BVDV isolates can be divided into cytopathic (cp) and non-cytopathic (ncp), with the ncp isolates being responsible for most natural infections and persistent fetal infections, and cp isolates constituting a minority, which are isolated almost exclusively from cattle with mucosal disease [6].

Cattle are natural hosts of BVDV, considered the major source of infection for pigs and other animal species [7,8]. Usually, positive pig herds for BVDV occur when cattle and pigs are raised on the same farm, and the direct contact between these animal species is considered the main source of BVDV transmission for pigs [7]. Infection caused by BVDV in pigs has been reported in China [9], the Netherlands [10], Brazil [11,12,13], Austria [14], Germany [15], Norway [16], Ireland [17], Denmark [18] and others. These data were found not only in domestic pigs but also in wild boars [19], which raise concerns about risk factors involved in BVDV infection, the clinical form of the disease, and the existence of accurate diagnostic tests. In Brazil, BVDV-1d was frequently reported in cattle [20]. Mósena et al. [11] states that by the phylogenetic analysis of sequenced samples collected from backyard pigs, classified as BVDV-1d and BVDV-2a, it is possible that one of the obtained sequences originated from contact between cattle and pigs.

It is known that all pestiviruses are genetically and antigenically related, and BVDV infection in pigs may be presented with a great variability of clinical signs [21]. Even though BVDV infections in pigs are not as problematic as Classical Swine Fever Virus (CSFV) infections, it is believed that distinguishing these two diseases could be difficult due to the similar clinical signs when considering low pathogenicity strains [22]. Reports of clinical signs associated with the infection consisting of anemia, delayed development, rough hair, polyarthritis, congenital tremors (CT), petechiae on the skin, diarrhea, conjunctivitis, and cyanosis [23]. Clinical signs similar to CSF, and sudden death [24] were also observed when the BVDV strain was isolated from both pigs and cattle from the same farm [24]. On the other hand, several recent studies with experimental infection did not report the presence of clinical signs of infection [25,26,27,28,29,30,31,32,33,34]. This may occur due to an inadequate level of viremia or a low virulence strain, biotype of the virus, host adaptation and/or route of inoculation [31,32,33,34,35]. A possible explanation is that cases in which BVDV infection-induced large numbers of lesions in adult pigs have been caused by viral strains that passed along previous adaptations in this species [23].

BVDV has a predilection for replication in defense cells, mainly lymphocytes, but it also infects monocytes and dendritic cells. As antigen presenters, dendritic cells play an important role in cellular immunity by initiating the nonspecific immune response against various pathogens [35]. Its infection promotes lysis of monocytes as a mechanism for evading the immune system, affecting the recognition and subsequent development of a specific immune-humoral response [36].

BVDV can contaminate cell cultures and fetal calf serum [37]. In countries that promote CSFV vaccination, the BVDV prevalence found in swine herds has been associated with the widespread use of live vaccines for Classical Swine Fever (CSF), which were produced with bovine sera from positive Chinese bovine herds [9]. Batches of live CSFV vaccines used in China confirmed five BVDV-contaminated samples out of 23 collected for testing [38].

Serological diagnosis by enzyme-linked immunosorbent assay (ELISA) is more efficient, cheaper, and faster than molecular techniques [37]. The neutralizing antibody titers in the serum of animals previously exposed to a Pestivirus member are usually medium to high regarding the homologous viral species, and low (or non-reactive) regarding other species [6]. Anti-BVDV antibodies were shown to be able to protect pigs against CSFV infection and the manifestation of clinical signs, even though the anti-CSFV antibody titers were low, which could hinder CSFV outbreaks in herds with a high prevalence of anti-BVDV antibodies [22]. The same condition could occur in the presence of anti-Border Disease Virus (BDV) antibodies, as cross-reactions could affect the transmission of CSFV and should be evaluated for an accurate diagnosis of a CSFV infection and for implementing specific surveillance protocols in cases of outbreaks [10].

Reverse transcription-polymerase chain reaction (RT-PCR) is widely used for detecting the viral agent for differential diagnosis [39], since samples of blood, milk, saliva, and tissue can be successfully tested [40], and can be stored for a long time with minimal losses [41]. Researchers have adapted numerous variations of PCR methods for the detection of infectious agents, using DNA templates as well as RNA templates after an RT step [42], which has enabled more accurate, sensible and specific diagnostics. Direct sequencing of the RT-PCR product for fragments of 5′UTR and N-terminal autoprotease (N^pro^) may also provide accurate differential diagnosis [19].

Given the antigenic and genetic similarity and the improvement in laboratory diagnostic methods, the comparison between results from recent and former studies should be cautious. Several studies that examine data collection in the field, as well as experimental infection with BVDV, have been conducted, and the results that contrast with the former data in the literature will be further discussed.

## 2. Data Collection from Backyard and Intensive Pig Herds

Aiming at collecting data on the occurrence of anti-BVDV antibodies in Brazilian swine herds, cross-sectional studies were carried out [12,13] in the backyard and intensive pig herds, respectively, located in the CSF-free zone of Brazil. For the first study, 56 pig herds from the northwest region of the state of São Paulo were evaluated, which are part of 11 municipalities. Blood samples were collected for serological testing by virus neutralization (VN), and titers higher than ten were considered positive. Out of the 360 serum samples, 4.72% (17/360) were reactive to BVDV in VN, which is 1.94% (7/360) reactive to BVDV-1 (Singer strain), with antibody titers ranging from 10 to 640, and 3.06% (11/360) reactive to BVDV-2 (VS-253 strain), with antibody titers ranging from 10 to 80, and only one reactive sample against both genotypes. Regarding herds, 27% (15/56) presented at least one animal positive to any of the genotypes. The prevalence of BVDV in bovine herds in the same region where this study was conducted was 56.49% [43], which may have resulted in the highest prevalence values of swine in the region when compared to previous reports [10]. Most of the farms evaluated in this study had cattle and pigs in close contact. As ruminants are the main source of BVDV infection for pigs [7,8], the prevalence of the disease in cattle herds is closely related to the presence of infections and influences the prevalence of the disease in pigs [9,10,44].

On the other hand, in a cross-sectional study carried out in 33 commercial pig herds, collected 1705 blood samples for analysis [13]. Samples were also tested by VN, and 5.34% (91/1705) of the samples were sero-reactive to BVDV with antibody titers ranging from 10 to 80. Of these, 3% (51/1075) were positive for reference strains of BVDV-1 (Singer strain) and 2.35% (40/1705) for reference strains of BVDV-2 (VS-253 strain), with 0.1% (2/1705) of samples with cross-reactions between both genotypes. Herds were sampled from 27 municipalities, located in seven Brazilian states, which are part of three different regions (South, Southeast, and Midwest). In 64% of herds (21/33), there was at least one positive sample for any of the BVDV genotypes in VN. As the presence of anti-BVDV antibodies in swine serum can lead to false-positive results in serological tests for the diagnosis of CSFV, the positive samples from both studies were sent for anti-CSFV antibodies detection, and were both negative. A survey carried out in The Netherlands [10] on commercial farm animals found a prevalence of 2.5% for gilts and 0.42% for finishing animals via ELISA. The difference in prevalence found in finishing animals from these two studies can be explained not only by the sensitivity of the techniques used but also by the different levels of biosecurity in the farms studied.

In a more recent study [11], swine sera were collected from 320 backyard pig herds in southern Brazil. Serum samples were tested by VN against BVDV-1a, -1b, and -2 strains, resulting in 4.2% (27/639) positive samples. Of those, 16 samples presented the highest titers against BVDV-1a (2 samples), BVDV-1b (5 samples), and BVDV-2 (9 samples). These studies confirm that ruminant Pestiviruses have been circulating in swine herds and must be considered in future Pestivirus control programs conducted in Brazil.

The low prevalence of BVDV in pigs was also found by other authors in Norway, Ireland, Denmark, and The Netherlands [16,17,18] (Table 1). Not only domestic pigs are a concern when it comes to Pestivirus infections since wild boar have been described as important reservoirs or transmitters of pathogens in nature due to their ability to reach long distances and transmit diseases to domestic swine. Other studies [45,46,47] have reported a low prevalence of BVDV in wild boar in Germany, the Czech Republic, and Eastern Serbia, respectively. Weber [19] was the first to detect BVDV RNA in wild boars’ blood samples, and Gatto [48] first reported the presence of anti-BVDV antibodies in Thayasuids. In general, the low prevalence of anti-BVDV antibodies in pigs may be associated not only with the level of interaction between pigs and ruminants but also with the host-pathogen specificity, which seems to be lower in pigs compared to cattle. In intensive pig farming, biosecurity measures reduce or eliminate the presence of some infectious agents. Some researchers [10,44] attributed the low prevalence of BVDV in swine herds in their studies to the high specialization of agriculture, in which interspecific contact was reduced due to single zootechnical breeding on the property.

In China, swine samples with clinical signs such as diarrhea, miscarriage, and death, between 2007 and 2010, were tested for BVDV by nested RT-PCR. Unlike that heretofore described, the observed prevalence was 23.1% in 2007, 27.7% in 2008, 33.6% in 2009, and 23.6% in 2010, showing a high prevalence of BVDV-1 infection [9] when compared to the abovementioned study. These numbers should be analyzed carefully since only samples from clinical cases compatible with BVDV infection were analyzed. As stated by these authors, the use of live vaccines against CSFV may also be directly related to this higher prevalence of BVDV in pig herds from China, since, in general, the prevalence of BVDV in pig herds is low.

## 3. Experimental Infection with BVDV in Pigs: Routes of Transmission and Disease Development

The lack of studies concerning the routes of transmission of BVDV between piglets highlighted the need to develop researches clarifying this information (Table 2). Three studies were separately conducted with three groups of two weaned piglets separated by isolation cabinets: the challenged, sentinel, and control groups. The isolation cabinets were arranged to allow only a specific route of transmission, namely airborne, nose-to-nose [25], and by back pond water [26]. Although all challenged piglets shed the virus and seroconverted, only transmission by the back-pond water was confirmed since sentinel animals also shed the virus and seroconverted in this study. An interesting fact regarding the viral shedding observed in these studies was the intermittent pattern of nasal shedding. Challenged piglets shed the virus between 5- and 24-days post-inoculation, intermittently, detected by RT-PCR of nasal swabs. In the challenged groups, clinical signs such as diarrhea, rough hair, and oculo-nasal discharge were observed about 15 dpi, when the piglets started seroconversion. Despite other pathogens that have not been searched for, these clinical signs may be suggestive of BVDV infection since it was not observed in piglets from the control group.

These studies proved that BVDV can infect weaned piglets, which shed the virus by the nasal route, presented clinical signs, and seroconverted. The presence of BVDV in nasal secretions indicates that pigs can be a source of infection for other animals, especially if piglets become infected by high virulent strains of BVDV. In the literature, BVDV shedding by a persistently infected boar has also been reported [49], with the detection of the virus in oropharyngeal fluid, urine, and semen.

Although BVDV infection in young pigs can occur without clinical manifestation of the disease [32], other authors have reported the appearance of reproductive problems in pregnant gilts, such as abortions, birth of small piglets, stillborn and congenital persistently infected animals (PI) [22]. In cattle, during acute infection, viremia and viral shedding are usually transient and at low titers, but even so, they can result in vertical transmission [50]. BVDV-2 infection can lead to the occurrence of fetal malformations in cattle; however, if the fetus survives the infection long enough, non-specific changes in maturation may occur in the lymphoid tissues [51]. In bovine herds infected by BVDV, several fetal malformations were described comprising cerebellar hypoplasia, poor myelinization of the spinal cord, hydrocephalus, microcephaly, retinal atrophy or dysplasia, and many others [6]. Diseases affecting myelin sheath formation or nerve synapses alter electrical impulses in neurons, which may lead to tremors [52].

Congenital infection of piglets born from gilts infected with BVDV has been reported in some studies [22,23], in which piglets died between the 2nd and the 16th week of life with signs similar to that of CSF, in addition to showing growth retardation. In a study in which pregnant sows were inoculated with BVDV on the 35th and 45th day of gestation by intrauterine and intranasal route, respectively, nine piglets born from females infected by the intrauterine route and five born from animals infected by intranasal route were born persistently infected [22]. Other studies have also shown transplacental infection, with the virus isolated in at least one of the fetuses [32,53,54].

Conversely, Pereira [27] inoculated BVDV-2 in groups of pregnant gilts at different stages of gestation and before artificial insemination (AI). Seroconversion and a transient viremia were detected, but reproductive losses and clinical signs of the disease in gilts and piglets were not observed. Other recent studies analyzed groups of gilts inoculated by BVDV-2 by oronasal [28,29] and intrauterine [28] routes on the 45th day of pregnancy, before the fetal immunocompetence period. No transplacental transmission was observed since piglets from oro-nasally inoculated gilts were born BVDV-free; no anti-BVDV antibodies were detected in piglets at birth but were acquired by colostral passive transfer. Congenital persistent infection was not observed since piglets did not shed BVDV at any moment. Rates of BVDV transmission between pigs under field conditions is very low, and under experimental conditions it would be even more limited [33].

Regarding intrauterine inoculation [28], piglets were born with no clinical signs of infection and no signs of hypomyelination or CT. Surprisingly, high anti-BVDV-2 antibody titers were found. Serological investigations in bovine fetuses experimentally inoculated with the virus also indicated the development of specific immune competence before the period already established in the literature [51]. The average period of seroconversion of the gilts challenged with the virus was 20 days [28], varying between 12 and 33 days [27]. Other studies have described BVDV inducing viremia seven days post-infection and seroconversion three weeks after experimental inoculation in pigs [55,56,57]. BVDV was discarded as an etiological agent of CT [28], differently from atypical porcine pestivirus (APPV), which was linked to CT-disease in experimental and natural infection conditions [58,59,60,61].

Understanding the role of BVDV in the reproductive system of boars is valuable information, considering that biotechnological procedures have the expressive potential of spreading diseases to free herds [62]. When it comes to boars, the presence of agents of the genus Pestivirus has been confirmed in porcine semen. Shedding of CSFV in porcine semen was the first to be reported under natural and experimental infection [63], as well as virus transmission to sows and fetuses by AI and transplacentally, respectively [64]. Recently, APPV was also detected in the semen and preputial fluid of naturally infected boars, with a high viral load in semen [65]. Experimental infection with BVDV-2 did not result in changes in the post-period of pre-inoculation in most of the seminal characteristics evaluated, and no viral shedding was detected in semen or preputial fluid, but lymphocytosis and monocytopenia were observed [30]. Considering a mild and transient viremia, the likelihood that the circulating virus in the blood reaches different organs was low. Also, the blood-testis barrier would decrease the chance of reaching semen, which may explain the absence of viral RNA detection in the reproductive tract of the inoculated boars [30]. A BVDV persistently infected boar presented viral shedding in the ejaculate, which contained no sperm cells [49]. Possibly, BVDV transmission by semen occurs in atypical cases of congenital persistent infection in pigs [49].

## 4. Final Consideration

The course of BVDV infection in pigs will depend on the virulence of the viral strain and the pig immune response [66] and may be limited [44]. Even so, the presence of the virus in the nasal secretions of infected animals demonstrated that pigs could act as a source of infection, thus facilitating the spread in the herd [26,27]. Although BVDV does not pose the same threat to pig herds as it poses to ruminants, it may lead to the development of a range of clinical signs and culminate in a serological cross-reaction with the CSFV, interfering negatively in classical swine fever monitoring and surveillance programs, and misleading diagnosis of the disease [10].

## Figures and Tables

**Table 1 viruses-12-00600-t001:** Bovine viral diarrhea virus prevalence in intensive pig farming, backyard pig herds, and wild boars observed by several authors in different regions worldwide.

References	Type of Pig Production	Animal Category	Number of Samples/Number of Herds	Type of Sample	Diagnostic Tool	Region	BVDV Prevalence
Loken et al., 1991	-	Adult pigs	1317/887	Serum	Virus neutralization	Norway	2.2%
Graham et al., 2001	-	-	660/46	Serum	ELISA*	Northern Ireland	0.15%
Loeffen et al., 2009	Intensive pig farming	Finishing pigs	1890/189	Serum	ELISA and Virus neutralization	The Netherlands	0.42%
Loeffen et al., 2009	Intensive pig farming	Sows	6020/616	Serum	ELISA and Virus neutralization	The Netherlands	2.5%
Deng et al., 2012	-	Pigs with clinical signs	511/11	Serum and tissue homogenate	Nested-RT-PCR	China (herds from 11 provinces)	26.8%
Gatto et al., 2017	Intensive pig farming		1705/33	Serum	Virus neutralization	Brazil (herds from six states)	5.34%
Almeida et al., 2017		Piglets and adults	360/56	Serum	Virus neutralization	Northwestern region of São Paulo State, Brazil	4.72%
Mósena et al., 2020	Backyard pig herds	Male and female animals 6–72 mo old	639/320	Serum	RT-PCR and virus neutralization	Rio Grande do Sul, Brazil	4.2%
Dahle et al., 1993	Wild boars	-	841 samples	Serum	Direct Neutralizing Peroxidase linked antibody assay (NPLA)	Northern Germany	0.83%
Sedlak et al., 2008	Wild boars	-	352 samples	Serum	ELISA	Czech Republic	1%
Milicevik et al., 2018	Wild boars	-	50 samples	Spleen	qRT-PCR	Eastern Serbia	8%
Weber et al., 2016	Farmed wild boars	-	40 samples	Lung	RT-PCR	Rio Grande do Sul, Brazil	2.5%
Gatto et al., 2020	Farmed white-lipped peccaries	-	72 samples	Serum	Virus neutralization	Midwest Brazil	1.38%

*ELISA: enzyme-linked immunosorbent assay.

**Table 2 viruses-12-00600-t002:** Results obtained in experimental inoculation studies of bovine viral diarrhea virus in swine.

References	BVDV Strain	Route of Inoculation	Animal Category	BVDV-Shedding	Viremia	Seroconversion (Dpi / % of Positive Pigs)	Consequences of Infection
Stewart et al., 1980	-	Intranasal-oral	Pregnant sows	-	7 dpi	21 dpi / 100%	Intrauterine infection in one litter, fewer fetuses than corpora lutea. Not observed in fetuses
Terpstra and Wensvoort, 1988	76/4 and 77/5 strains	Natural infection	Sows and piglets	-	-	35 dpi / (NA)	Piglets with clinical signs similar to CSF
Paton and Done, 1994	91/1 and 87/6 strains	Intrauterine	Pregnant sows	only congenital persistently infected fetus, for 2.5 years	4–6 dpi in fetuses	variable in piglets, but all sows seroconverted	Intrauterine infection, some fetuses with no clinical signs, some with persistent congenital infection
Terpstra and Wensvoort, 1997	Van Ee, Appel and Toering	Natural infection	Piglet	urine, oropharyngeal fluids, and semen of persistently infected boar	Observed	30 dpi - 8 mo / (NA)	Congenital persistent infection, leukopenia, viral replication in several organs, clinical signs resembling chronic CSF
Kulcsar et al., 2001	Oregon C24V strain	Intranasal and subcutaneous	Pregnant sows	Not observed	-	28 dpi / 100%	No clinical signs on sows, birth of weak piglets with ruffled hair coat, splay leg, trembling, myoclonia, diarrhea, fever, death
Makoschey et al., 2002	BVDV-2	-	Piglets	-	-	Observed / 100%	Inapparent infection with a slight increase in body temperature in some piglets, viral replication in cells/organs. Slight leukopenia and/or thrombocytopenia
Walz et al., 2004	BVDV-1 ncp	Intranasal inoculation	Pregnant gilts	-	5–7 dpi	21 dpi / 100%	No clinical signs observed in gilts or piglets; transplacental infection occurred in only one fetus from one gilt. No antibodies were found in piglets
Wieringa- Jelsma et al., 2006	St. Oedenrode strain	Intranasal inoculation	Weaned pigs	2–11 dpi	7 dpi	21 dpi / 100%	Not observed
Dos Santos et al., 2017	BVDV-1 (Singer) cp	Oronasal inoculation	Weaned piglets	10–25 dpi	Not observed	25 dpi / 62.5%	Diarrhea, rough hair, nasal discharge
Nascimento et al., 2017	BVDV-1 (Singer) cp	Oronasal inoculation	Weaned piglets	5–24 dpi	Not observed	25 dpi / 100%	Diarrhea, shivering, nasal discharge
Mechler et al., 2018	BVDV-2 (VS-260) ncp	Oronasal and intrauterine inoculation	Pregnant gilts	No	Not observed	20 dpi / 100%	Not observed
Pereira et al., 2018	BVDV-2 (VS-253) cp	Oronasal inoculation	Pregnant gilts	6–24 dpi	3–12 dpi	12–33 dpi / 70%	Thrombocytopenia
Gomes et al., 2019	BVDV-2 (VS 260) ncp	Oronasal inoculation	Pregnant gilts	No	Not observed	20 dpi / 100%	Not observed
Storino et al., 2020	BVDV-2 (LVB 16557/15) ncp	Oronasal, intramuscular and intravenous inoculation	Boars	No	Not observed	20–40 dpi / 12.5%	Lymphocytosis and monocytopenia

Dpi: days post-inoculation; cp: cytopathic; ncp: non-cytopathic.; NA = not applicable.

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
