# Peer review of "Bovine Viral Diarrhea Virus: Recent Findings about Its Occurrence in Pigs"

_viruses, 2020, doi:10.3390/v12060600_

Round 1
Reviewer 1 Report
Overall, the manuscript needs help in terms of focus and organization. That is, it touches on a lot of topics, but the logic for the order of presentation is difficult to follow and the information is too superficial. For example, the abstract rightly says that "the presence of antibodies anti-BVDV in swine herds may interfere in the surveillance of CSF", but this topic is not fleshed out in the body of the reivew. Of course, there is an massive body of literature on BVDV in pigs, cross reactions with CSFV tests, etc. It is challenging to integrate this enormous body of information into a short review. My suggestion would be to narrow the topic, e.g., Update on BVDV Infections in Pigs. The "Update" could be on current estimates of BVDV infections in commercial pigs; recent experimental data; contemporary diagnostics and cross reactivity among pestiviruses.
Author Response
Dear Viruses Editor and Reviewers,
We are very grateful for the opportunity of having our manuscript considered for publication in this important journal. We have accepted and corrected all items suggested by the Reviewers, and provided a better explanation of specific points for clarifying our idea.
We understand that there is plenty information regarding BVDV infection on pigs, and we wanted to bring data from recent experiments that contrast with the former ones. We have made a suggestion to narrow the manuscript title, which should better comprise the content of the work.
We are extremely thankful for all the suggestions, because we know that all of them were very important for improving the quality of this work. We also stated the changes made to the manuscript in response to the respective comment. Thank you for the opportunity to improve our work, we hope to receive a positive response.
Reviewer 1 - Comments |
Correction |
Overall, the manuscript needs help in terms of focus and organization. That is, it touches on a lot of topics, but the logic for the order of presentation is difficult to follow and the information is too superficial. For example, the abstract rightly says that "the presence of antibodies anti-BVDV in swine herds may interfere in the surveillance of CSF", but this topic is not fleshed out in the body of the reivew. Of course, there is an massive body of literature on BVDV in pigs, cross reactions with CSFV tests, etc. It is challenging to integrate this enormous body of information into a short review. My suggestion would be to narrow the topic, e.g., Update on BVDV Infections in Pigs. The "Update" could be on current estimates of BVDV infections in commercial pigs; recent experimental data; contemporary diagnostics and cross reactivity among pestiviruses. |
We would like to thank the Reviewer 1 for the comments. We tried to rewrite and change the order of some sentences to make the comprehension more clear, and we also changed the heading of the topic as suggested. We have made a suggestion to narrow the manuscript title, which should better comprise the content of the work |

Reviewer 2 Report
The manuscript describes the review article of BVDV infections in pigs.
In general all important topics are included but not clearly and comprehensive presented for reader in manuscript.
The statements from the literature are presented only as partial observation, not completely ''state of the art'' foe review article. This is difficult topics, but should be presented more comprehensive.
Some concerns:
Chapter 1 Update and general informations...
Lines 27-33 and 34-41: Old and new classification are presented one afer another-this should be explained properly
Lines 45-48: this statement can be moved down among reported countries (line 50).
Lines 54-69: Rather old references were used here: In general differential diagnosis of pestivirus antibodies can be obtained with VN test and using diffrent reference strains of CSFV, BVDV and Border disease virus, including field strains. The higher titer is obtained for homologus strain...
General data about cross reactivity in VN is missing
Lines 77-78: need revision, not clear
Line 79: Serological diagnostic (by ELISA test?)...
Lines 85; PSC vaccine (PSC?)
Line 88: RT-PCR -not celar (RT-reverse trascription or real-time PCR)
Chapter 2: Data collection...
Line 100: it is not clear the meaning of technified and non-technified pig herds
Line 104: not celar which strains were used for VN (reference of field strains)
Lines 103-107: nothing about titer of detected antibodies, also data about cohabitate of pigs and cattle is important.
Lines 112-118: What about Border disese virus antibodies
Table 2: Important data is mnissing-% of seroconverted pigs in different experimental infections, also data about titers for homologous strain and heterologos strains, if available should be presented
Author Response
Dear Viruses Editor and Reviewers,
We are very grateful for the opportunity of having our manuscript considered for publication in this important journal. We have accepted and corrected all items suggested by the Reviewers, and provided a better explanation of specific points for clarifying our idea.
We understand that there is plenty information regarding BVDV infection on pigs, and we wanted to bring data from recent experiments that contrast with the former ones. We have made a suggestion to narrow the manuscript title, which should better comprise the content of the work.
We are extremely thankful for all the suggestions, because we know that all of them were very important for improving the quality of this work. We also stated the changes made to the manuscript in response to the respective comment. Thank you for the opportunity to improve our work, we hope to receive a positive response.
Reviewer 2 - Comments |
Correction |
Lines 27-33 and 34-41: Old and new classification are presented one after another-this should be explained properly |
We have change the order of the statements, and now we believe that information is presented in a clearer way to the readers. |
Lines 45-48: this statement can be moved down among reported countries (line 50).
|
Done as requested |
Lines 54-69: Rather old references were used here: In general differential diagnosis of pestivirus antibodies can be obtained with VN test and using diffrent reference strains of CSFV, BVDV and Border disease virus, including field strains. The higher titer is obtained for homologus strain...
|
A: Yes, we agree. We have removed the old information, and at the second last paragraph, a correct information regarding differential diagnosis is presented.
|
General data about cross reactivity in VN is missing
|
A: This information is briefly presented at the second last paragraph of the first topic.
|
Lines 77-78: need revision, not clear
|
We chose to remove this sentence as it was not necessary to the comprehension of the previous sentence. |
Line 79: Serological diagnostic (by ELISA test?)...
|
Yes, by ELISA. The information was corrected in the text.
|
Lines 85; PSC vaccine (PSC?)
|
There was a mistranslation problem, and we apologize for that. The correct is CSF. |
Line 88: RT-PCR -not celar (RT-reverse trascription or real-time PCR)
|
Reverse transcription PCR, the term was added in the text. |
Chapter 2: Data collection... Line 100: it is not clear the meaning of technified and non-technified pig herds
|
Non-technified are backyard creations, and technified are intensive pig farming. We changed the terms to a better understanding
|
Line 104: not clear which strains were used for VN (reference of field strains)
|
Reference strains were added to the text |
Lines 103-107: nothing about titer of detected antibodies, also data about cohabitate of pigs and cattle is important. |
Antibody titers found in the studies were included in the manuscript, as well as the information about the presence of cattle in the backyard creations. |
Lines 112-118: What about Border disese virus antibodies
|
This study did not evaluate border disease antibodies. |
Table 2: Important data is mnissing- % of seroconverted pigs in different experimental infections, also data about titers for homologous strain and heterologos strains, if available should be presented
|
The percentage of seroconverted pigs of each study was added to the table, but information regarding titers for homologous strains were not available.
|

Reviewer 3 Report
The purpose of this generally well written manuscript is to review the current published literature on BVDV in swine. Controversy exists over the importance of BVDV infections swine that have been reported since the 1960s, and several studies have been published on this subject. These studies reported differing results about the ability of BVDV to cause infection and clinical signs in pigs. Clinical signs can range from completely asymptomatic, to signs of anemia, delayed development, rough hair coats, diarrhea, and others. Seroprevalence ranges dramatically in published reports, from less than 1% to 64% in different countries using different diagnostic techniques. BVDV infections in swine have been reported from both natural and experimental infections with a variety of methods of inoculation. As written, the manuscript focuses too much on the work by this group of authors, which likely has led to the conclusion that BVDV is not a threat to pigs – this is not the case based on other studies. Therefore the authors should discuss the conflicting results from previous studies, rather than omitting some of that work. While several improvements should be made to this manuscript, a review of the topic would be a valuable addition to the literature.
Specific comments:
- In the abstract, authors state “Although BVDV does not pose a threat to pigs…”. However, many reports have demonstrated both experimental and natural infections resulting in clinical signs, this statement is not accurate. BVDV may not pose a similar threat to pigs as it does to ruminants, but there are certainly possible health effects. This statement should be revised…
- As in comment 1, the same strong statement is made in the final considerations on line 233. As clinical signs can be observed under certain circumstances, we can’t definitively say that BVDV is not a threat to pig herds.
- The abstract (and throughout) states that “this study aimed…” Please remove such language. The authors did not perform a study but wrote a review article.
- Line 67- The reason for an absence of clinical signs in the studies conducted by your research group was postulated to only occur due to inadequate viremia. Is it not feasible that a low virulence strain in that species, serotype or biotype of the virus, host adaptation, or route of inoculation also be potential explanations for the lack of clinical signs?
- Line 85- PSC is utilized, but no definition of the term is provided previously.
- Line 86- PCR is utilized, but no definition of the term is provided previously.
- Line 144- interspecies to interspecific
- The entire manuscript is too focused on the work by this group of researchers. An example is the paragraph starting in line 157. Why did the authors omit work by Wieringa-Jelsma 2006, Paton and Done 1994, reports of clinical disease following use of contaminated vaccines etc. Much work has been done with BVDV in pigs and this previous work should be compared and contrasted, rather than omitted. Some of the other work contrast the research by this group of authors, and it is critical that it be discussed.
- Line 205. In the cited work, pigs were inoculated on day 45 of gestation, which is likely outside of the window of susceptibility for creation of PI piglets. This should be discussed.
- Section 3, lines 214-228 discuss of absence or presence of virus in semen in infected boars. As referenced #49, Terpstra and Wensvoort detail a persistently infected boar piglet that had virus present in the ejaculate, but no sperm were present. As no sperm was present in the ejaculate in this boar, it is speculative if horizontal transfer through insemination could occur, but vertical transmission would be possible.
- Line 231- “maybe” needs to be changed to “may be”
Author Response
Dear Viruses Editor and Reviewers,
We are very grateful for the opportunity of having our manuscript considered for publication in this important journal. We have accepted and corrected all items suggested by the Reviewers, and provided a better explanation of specific points for clarifying our idea.
We understand that there is plenty information regarding BVDV infection on pigs, and we wanted to bring data from recent experiments that contrast with the former ones. We have made a suggestion to narrow the manuscript title, which should better comprise the content of the work.
We are extremely thankful for all the suggestions, because we know that all of them were very important for improving the quality of this work. We also stated the changes made to the manuscript in response to the respective comment. Thank you for the opportunity to improve our work, we hope to receive a positive response.
Reviewer 3 - Comments |
Correction |
In the abstract, authors state “Although BVDV does not pose a threat to pigs…”. However, many reports have demonstrated both experimental and natural infections resulting in clinical signs, this statement is not accurate. BVDV may not pose a similar threat to pigs as it does to ruminants, but there are certainly possible health effects. This statement should be revised… As in comment 1, the same strong statement is made in the final considerations on line 233. As clinical signs can be observed under certain circumstances, we can’t definitively say that BVDV is not a threat to pig herds. |
We agree that the way the statements were presented, the information was too limited and the affirmation was very strong. The statements referred in comments 1 and 2 were revised and reformulated as suggested. |
The abstract (and throughout) states that “this study aimed…” Please remove such language. The authors did not perform a study but wrote a review article |
The term was changed where necessary.
|
Line 67- The reason for an absence of clinical signs in the studies conducted by your research group was postulated to only occur due to inadequate viremia. Is it not feasible that a low virulence strain in that species, serotype or biotype of the virus, host adaptation, or route of inoculation also be potential explanations for the lack of clinical signs? |
Yes, we totally agree with this comment. We thought that the next sentence would clear this information, but now we have reformulated the sentence. “This may occur due to an inadequate level of viremia, a low virulence strain, biotype of the virus, host adaptation, or route of inoculation [31-35] ” |
Line 85- PSC is utilized, but no definition of the term is provided previously.
|
This term was mistranslated. The correct is CSF and it was changed in the text.
|
Line 86- PCR is utilized, but no definition of the term is provided previously |
Definition of PCR was properly added |
The entire manuscript is too focused on the work by this group of researchers. An example is the paragraph starting in line 157. Why did the authors omit work by Wieringa-Jelsma 2006, Paton and Done 1994, reports of clinical disease following use of contaminated vaccines etc. Much work has been done with BVDV in pigs and this previous work should be compared and contrasted, rather than omitted. Some of the other work contrast the research by this group of authors, and it is critical that it be discussed. |
With all the respect, we absolutely did not want to omit information regarding the works cited by the reviewer. Both studies were already cited in the text (references #22 and #33), and data are also present in Table 2, which compares the findings of experimental BVDV infection in several studies. Data regarding clinical disease by the use of contaminated vaccine was already cited in the text as well (references #11 and #38). We understand that there is a world of information regarding BVDV infection on pigs, and the objective of writing this review was to present recent findings about it, comparing with the older ones. |
Line 205. In the cited work, pigs were inoculated on day 45 of gestation, which is likely outside of the window of susceptibility for creation of PI piglets. This should be discussed |
The inoculation occurred in the middle third of gestation, prior to the fetal immunocompetence, which is reported to occur near the 70th day of gestation. For that reason it is believed that BVDV infection in this period may lead to congenital persistent infection in piglets.
|
Section 3, lines 214-228 discuss of absence or presence of virus in semen in infected boars. As referenced #49, Terpstra and Wensvoort detail a persistently infected boar piglet that had virus present in the ejaculate, but no sperm were present. As no sperm was present in the ejaculate in this boar, it is speculative if horizontal transfer through insemination could occur, but vertical transmission would be possible.
|
We agree that the way that sentences were written there was a misunderstanding of the real idea that we wanted to pass. This point was restructured. |
Line 231- “maybe” needs to be changed to “may be” |
The correction was made as requested. |

Round 2
Reviewer 2 Report
The manuscript was significantly improved after revision and is now suitable for poblication. However, som minore remarks need to be implemented before publication.
- Line 20: correct ''two dots'' at the end of sentece.
- Line 59: Classical Swine Fever (CSF) already explained in line 55: correct into ''CSF''.
- Line 84: Revise this sentence, specific RT-PCR methods are nowdays widely used for detection of viruses (PCR is not any more ''good alternative''), also direct sequencing from PCR product (eg in 5'NTR) should be metioned as an option for differentilal diagnosis between different pestivirus strains.
- Line 94: Into chapter 2; Add also somewhere what is the titer, which is consider positive in VNT for BVD, when you are using VNT for differentioal disagnosis.
- Also few sentences should bee added about possible effect of Border disese virus infection and other atipical pestiviruses for differential diagnosis in infected herds (maybe for Brasil this is not so important, but for other part of the world should be considered as an option, if different animal species are keept close together in the same holding)
- Line 100: I will suggest use rether (Singer strain) instead only (Singer).
- Line 101: I will suggest use (VS-253 strain) ....
- Line 112: change into (Singer strain)
- Line 113: change into (VS-253 strain)
- Line 119: I will suggest to use ''The Netherlands'' instead of Holland
- Line 120: Check aggain this sentence, is not clear...for finishing animals ELISA.
- Line 130: use ''The Netherlans''
- Line 134: Check anggain this sentence, if no exact data is presented in sentence, two time mentioned ''Germany'' is not necessary.
- Line 200: delete ''(PI)'', here is enough ''congenital persistent infection''
- Line 204: explain the meaning of ''CT''- first mentioned
- Line 210: explain the meaning of APPV-first mentioned
- Line 217: APPV is enough
I hope that my remarks are acceptable for you and will be considered as valuable and implemeted into reviesed version to improve manuscript before publication.
Author Response
Reviewer 2 - Comments |
Correction |
Line 20: correct ''two dots'' at the end of sentence. |
This mistake was corrected. |
Line 59: Classical Swine Fever (CSF) already explained in line 55: correct into ''CSF''.
|
This information was corrected.
|
Line 84: Revise this sentence, specific RT-PCR methods are nowdays widely used for detection of viruses (PCR is not any more ''good alternative''), also direct sequencing from PCR product (eg in 5'NTR) should be metioned as an option for differentilal diagnosis between different pestivirus strains.
|
This information was corrected as requested. “Direct sequencing of the RT-PCR product for fragments of 5’UTR and N terminal autoprotease (Npro) may also provide accurate differential diagnosis [20]. “
|
Line 94: Into chapter 2; Add also somewhere what is the titer, which is consider positive in VNT for BVD, when you are using VNT for differentioal disagnosis.
|
Titers were considered positive when higher thatn 10, and this information was added in the text.
|
Also few sentences should bee added about possible effect of Border disese virus infection and other atipical pestiviruses for differential diagnosis in infected herds (maybe for Brasil this is not so important, but for other part of the world should be considered as an option, if different animal species are keept close together in the same holding)
|
An statement was added as suggested: “The same condition could occur in the presence of anti-BDV antibodies, as cross-reactions could affect the transmission of CSFV and should be evaluated for an accurate diagnostic of a CSFV infection and for implementing specific surveillance protocols in cases of outbreaks [12].”
|
Line 100: I will suggest use rether (Singer strain) instead only (Singer).
Line 101: I will suggest use (VS-253 strain)
Line 112: change into (Singer strain)
Line 113: change into (VS-253 strain)
Line 119: I will suggest to use ''The Netherlands'' instead of Holland
|
All of the items were corrected in the text. |
Line 120: Check aggain this sentence, is not clear...for finishing animals ELISA.
|
Probably during the last correction, the preposition “by” was omitted. We have corrected it to “by ELISA”.
|
Line 130: use ''The Netherlans''
|
Changed
|
Line 134: Check anggain this sentence, if no exact data is presented in sentence, two time mentioned ''Germany'' is not necessary.
|
We apologize for that mistake, the correct is Germany, Czeck Republic and Eastern Serbia.
|
Line 200: delete ''(PI)'', here is enough ''congenital persistent infection''
|
Deleted |
Line 204: explain the meaning of ''CT''- first mentioned
|
Congenital Tremor (CT) was firstly mentioned in line 58.
|
Line 210: explain the meaning of APPV-first mentioned
|
Atypical Porcine Pestivirus is now mentioned before the APPV.
|
Line 217: APPV is enough
|
Corrected.
|
